# A Study on the Transient Process of Contact-Mode Triboelectric Nanogenerators

**DOI:** 10.3390/mi16091070

**Published:** 2025-09-22

**Authors:** Shengyao Zhang, Hongchun Luo, Ru Zhang, Shun Ye, Haoyu Wei, Zhiqiang Zeng, Futi Liu, Guiyu Zhou

**Affiliations:** 1College of Mathematics and Physics, Yibin University, Yibin 644007, China; zhangshengyao@yibinu.edu.cn (S.Z.); 2010105003@yibinu.edu.cn (Z.Z.); 2Department of Civil Engineering, Hangzhou City University, Hangzhou 310015, China; 3College of Civil Engineering and Architecture, Zhejiang University, Hangzhou 310058, China; 4College of Electronic Information and Engineering, Yibin University, Yibin 644007, China

**Keywords:** triboelectric nanogenerator, transient process, steady state, Fourier expansion

## Abstract

In the research of triboelectric nanogenerators (TENGs), most attention has been paid to material modification, structural design, and power management. Little study has been performed on the transient process of TENGs, although the capacitance characteristics of TENGs are well known. In this work, the transient process of contact-mode TENGs was studied by the infinite-plate model and verified by experimental tests. The results showed that TENGs exhibited a much higher output in the transient process than that in the steady state. Within the transient process, the transfer charge gradually grew to a maximum value, while the output current and power decreased. A formula to calculate the duration of the transient process was derived by Fourier expansion. This work also demonstrated an interesting transformation process of the *Q*-*V* curve in the transient process. Furthermore, the transient phenomenon was verified clearly in a contact-mode TENG sample fabricated by copper and polytetrafluoroethylene (PTFE) films through experimental tests. These results are useful for performance optimization of TENGs in applications.

## 1. Introduction

With the rapid development of modern society, the huge energy requirement, particularly for artificial intelligence, becomes more challenging because conventional power sources pollute the environment greatly, meaning green energy sources are highly necessary. As one of the green energy technologies, triboelectric nanogenerators (TENGs) have attracted more and more attention in recent years due to their potential in various engineering applications including energy harvesting and self-powered sensing [1,2]. The triboelectrification process was a well-known phenomenon even thousands of years ago. After friction between two materials, some charges, called triboelectric charges, will transfer from one material to the other. If the electrodes attached to the two materials connect with each other, some free charges, called transfer charges, will accumulate on the electrodes due to electrostatic induction. When relative motion between the two materials happens, transfer charge will flow between the two electrodes and induce an electric current, and mechanical energy can be converted into electric energy for applications. TENGs exhibit many advantages over conventional energy-harvesting techniques, such as a simple structure, low cost, and high output power. Today, many efforts have been devoted to the study of triboelectric material modification, novel structure/mechanism design, power management, and proposed applications [3,4,5,6,7,8,9,10,11]. However, these studies are concerned only with the steady state of TENGs, and little attention has been paid to the transient process of TENGs at their working start, although the capacitance characteristics of TENGs are well known. Considering the importance of the transient process in circuit analysis theory for LC and RLC circuits, it seems that the transient process should also be studied carefully for TENGs; otherwise, important information supporting the performance optimization of TENGs may be ignored. To study the transient process, theoretical modeling is a very useful and important method for obtaining an in-depth understanding of the working mechanisms of TENGs. Many models have been established for the performance analysis of TENGS. Niu et al. established an infinite-plate model for contact-mode TENGs [12]. This theory is able to depict the intrinsic physical nature of TENGs as simply as possible and has become a basis for the following theoretical and experimental works on TENGs [13,14,15]. While an assumption of infinite large contact plates is employed in Niu’s model, the following studies try to utilize the concept of a distance-dependent electrical field (DDEF) to consider the edge effects in TENGs with finite sizes in order to obtain more accurate results [16,17,18,19,20,21,22,23]. Unfortunately, all the solutions from the following models are so complex in expression that they are not convenient for engineering applications, which usually prefer a simple rule of thumb, and they do not improve precision dramatically. Overall, compared with other models, the infinite-plate model is the most convenient, which can estimate the performance of TENGs with satisfactory accuracy.

In this work, by studying the time dependence of transfer charge, current, and power, the transient process of contact-mode TENGs at the start time of movement was studied both theoretically and experimentally. A formula for calculating the duration of the transient process was obtained by Fourier expansion in the theoretical analysis. The results showed that the output performance of TENGs was much higher in the transient state than that in the steady state, particularly for large load resistances. The results also showed an interesting transformation process of the *Q*-*V* curve during the transient process. Furthermore, the transient phenomenon was observed clearly in a contact-mode TENG sample fabricated by copper and PTFE films through experimental tests under large load resistance, which verified the theoretical predictions.

## 2. Theoretical Basis

A typical TENG model structure of the contact mode is shown in Figure 1, where a dielectric material with thickness (*d*) and relative dielectric permittivity ($\varepsilon_{r}$) is attached on a metal electrode while another metal electrode is placed above the dielectric material. An external load (*R*) is connected to the two electrodes to collect output power (*P*) from the TENGs. When the upper electrode contacts with the dielectric material, triboelectrification occurs and a number of triboelectric charges with surface density ($\sigma_{T}$) will transfer between the contact pair due to the difference in working potentials of the two materials. After that, if the upper electrode begins to move relative to the dielectric material, it will trigger the movement of transfer charge (*Q*_c_), which is the accumulated free charge on the bottom electrode from one electrode to another, to introduce an electric current into the external load. The governing equation to describe the variation in *Q*_c_ with time (*t*) is(1)$$R\frac{dQ_{c}}{dt}+\frac{d_{0}+x\left(t\right)}{S\varepsilon_{0}}Q_{c}=\frac{x\left(t\right)}{\varepsilon_{0}}\sigma_{T},$$
where $\varepsilon_{0}$ is the permittivity of air, *S* is the area of the dielectric material and electrodes, *x*(*t*) is the position of upper electrode relative to the dielectric material, and $d_{0}=d/\varepsilon_{r}$ represents the effective thickness of the dielectric layer.

The analytical solution obtained in this work by considering initial condition $Q_{c}\left(t=0\right)=0$ is expressed as the following form(2)$$Q_{c}=\left(\int_{0}^{t}\frac{\sigma_{T}x\left(t^{\prime }\right)}{R\varepsilon_{0}}e^{\frac{1}{RS\varepsilon_{0}}\left(d_{0}t^{\prime }+F\left(t^{\prime }\right)\right)}dt^{\prime }\right)e^{-\frac{1}{RS\varepsilon_{0}}\left(d_{0}t+F\left(t\right)\right)}$$
where $F\left(t\right)=\int x\left(t\right)dt$. From Equation (2), the output current and voltage of the TENGs can be calculated as(3)$$I=\frac{dQ_{c}}{dt}=\frac{\sigma_{T}x\left(t\right)}{R\varepsilon_{0}}-\frac{d_{0}+x\left(t\right)}{RS\varepsilon_{0}}\left(\int_{0}^{t}\frac{\sigma_{T}x\left(t^{\prime }\right)}{R\varepsilon_{0}}e^{\frac{1}{RS\varepsilon_{0}}\left(d_{0}t^{\prime }+F\left(t^{\prime }\right)\right)}dt^{\prime }\right)e^{-\frac{1}{RS\varepsilon_{0}}\left(d_{0}t+F\left(t\right)\right)}$$
and(4)$$V=RI=\frac{\sigma_{T}x\left(t\right)}{\varepsilon_{0}}-\frac{d_{0}+x\left(t\right)}{S\varepsilon_{0}}\left(\int_{0}^{t}\frac{\sigma_{T}x\left(t^{\prime }\right)}{R\varepsilon_{0}}e^{\frac{1}{RS\varepsilon_{0}}\left(d_{0}t^{\prime }+F\left(t^{\prime }\right)\right)}dt^{\prime }\right)e^{-\frac{1}{RS\varepsilon_{0}}\left(d_{0}t+F\left(t\right)\right)}$$
respectively. The instantaneous output power of the TENGs can be calculated by P=VI. It should be noted that the output characteristics of the TENGs are dependent on the motion mode of the upper electrode, which is described by *x*(*t*). To study the transient process, the motion mode of TENGs is chosen as the sinusoidal motion(5)$$x\left(t\right)=\frac{x_{m}}{2}\left(1-cos2\pi ft\right)$$
where $x_{m}$ is the maximum displacement and *f* is motion frequency. Substitution of Equation (5) into Equations (2)–(4) could give the time dependence of charge, current, voltage, and power. However, due to the difficulty of integration in these equations, the expression of charge *Q*_c_ cannot be obtained analytically. To divide the expression of *Q*_c_ into a sum of the transient-state term and steady-state term, Fourier expansion is performed:(6)$$e^{-\frac{x_{m}}{2\omega RS\varepsilon_{0}}sin2\pi ft}=\frac{a_{0}}{2}+\sum_{l=1}^{\infty }\left(a_{l}cos2\pi lft+b_{l}sin2\pi lft\right)$$
is adopted in the calculation in Equation (2), where(7)$$\begin{array}{l} a_{l}=2f\int_{-\frac{1}{2f}}^{\frac{1}{2f}}e^{-\frac{x_{m}}{4\pi fRS\varepsilon_{0}}\sin2\pi ft}\cos2\pi lftdt,\;\;l=0,1,2,\ldots \\ b_{l}=2f\int_{-\frac{1}{2f}}^{\frac{1}{2f}}e^{-\frac{x_{m}}{4\pi fRS\varepsilon_{0}}\sin2\pi ft}\sin2\pi lftdt,\;\;l=1,2,\ldots \end{array}$$
Then, *Q*_c_ can be expressed as(8)$$Q_{c}=A\left(t\right)e^{\frac{x_{m}}{4\pi fRS\varepsilon_{0}}sin2\pi ft}+Be^{-\left(\frac{d_{eff}}{RS\varepsilon_{0}}+\frac{x_{m}}{2RS\varepsilon_{0}}\right)t+\frac{x_{m}}{4\pi fRS\varepsilon_{0}}sin2\pi ft},$$
where $A\left(t\right)$ is a function of time, and *B* is a constant. The detailed forms of $A\left(t\right)$ and *B* are illustrated in Appendix A. It is clear that in Equation (8), the first term represents the steady state of TENGs, while the second term represents the transient state, because the amplitude decays with the time exponentially. Therefore, if the time of the transient process td is defined as the time needed for the amplitude to reduce to 1/*e* of its initial value, then the lasting time of the transient process of TENGs can be calculated by the following formula:(9)$$t_{d}=\frac{2RS\varepsilon_{0}}{2d_{eff}+x_{m}}.$$
This formula illustrates that the duration of the transient process is linearly dependent on the load resistance and electrode area and decreases with increasing $d_{eff}$ and $x_{m}$.

## 3. Results and Discussion

By substituting the data in Table 1 into Equations (2), (3), and (5), the results are obtained, as shown in Figure 2 below. Figure 2 presents the time dependence of transfer charge, current, and power for different load resistances. Figure 2a,d,g,j show that the amount of transfer charge increases gradually with time in the transient processes until reaching a maximum value of ~55.9 nC and then begins to vary sinusoidally in a steady state no matter what value of the load is being taken. The maximum value of transfer charge is determined by the amount of triboelectric charge generated in triboelectrification. It is also illustrated that the amplitude of transfer charge in the steady state decreases with the increase in load resistance. As shown in Figure 2j, the amplitude of variation reduces even to 0.01 nC when the load value increases to 1 GΩ. The reason underlying this phenomenon is that when the resistance becomes large, the flow of transfer charge accumulated on the electrode will be blocked by the resistance. This means that the movement of the transfer charge from one electrode to another becomes more difficult for a larger resistance. Therefore, the variation in transfer charge will be reduced with the increase in the load, resulting in a shrinkage of these charge curves. Figure 2b,e,h,k show the time dependence of electric current in the transient process for different loads. From these figures, it is clear that the amplitude of current decreases with time in the transient process until reaching a sinusoidal steady state. It should be noted that the lasting time of the transient process is related to the resistance of the load. A larger load resistance will lead to a longer transient time. As shown in Figure 2b,k, it takes just one or two time periods for the TENGs to reach the steady state when the load value is smaller than 100 MΩ, while for large loads (≥1 GΩ), up to ten or even more time periods are required for TENGs to reach the steady state. The time of the transition process (*t*_d_) for various load resistances is calculated from Equation (9) and shown in Figure 2a,d,g,j for reference. Figure 2c,f,i,l show that the amplitude of output power decreases with time in the transient process and decreases more slowly when the load is larger. From these figures, it can be found that with the increase in load, the current will be reduced, and the same trend is illustrated for power in the steady state. The reason for this is also that the large resistance increases the difficulty for flow of transfer charge in the circuit, which limits the variation in transfer charge and thus decreases the current and power.

Another way to study the transient process of TENGs is according to the curves describing the relationship between the transfer charge (*Q*_c_) and output voltage (*V*) of TENGs, namely, *Q*-*V* curves, which can be obtained from Equations (2), (4), and (5). Figure 3 illustrates the variation in *Q*-*V* curves with time during the transient process. Figure 3a,b plot the variation in *Q*-*V* curves within a time range of 0~100 ms for different loads. It can be found that the *Q*-*V* curves in the transient process form a sequence of open loops until a closed loop (corresponding to the steady state) comes into being. Each loop represents a time period, and the number of loops is increased with the increase in load resistance. This means the transient process time is increased with resistance. These observations are consistent with previous conclusions from Figure 2. It should be noted that the tail of *Q*-*V* curves in Figure 3b is enlarged in the upright subset figure for a clear view because the curves are too dense there. To study the *Q*-*V* curves more closely, four loops corresponding to different time periods are selected at a load resistance of 1 GΩ and replotted in Figure 3c–f, respectively. It is shown that *Q*-*V* curves will gradually form a closed loop from an initial open shape in the transient process.

## 4. Experimental Verification and Discussion

To observe the transient process in experiments, a sample of contact-mode TENGs was fabricated for tests. The sample consisted of two parts, as shown in Figure 4a. In Figure 4a, the left part is a copper (Cu) layer glued onto a rigid acrylic substrate, which was used as the upper moving electrode, as shown in Figure 1, and the right part is a dielectric polytetrafluoroethylene (PTFE) layer attached to a copper layer, which was used as the fixed bottom electrode, as shown in Figure 1. The PTFE-attached Cu layer in the right part was also glued onto an acrylic substrate. The thicknesses of the Cu and PTFE layers were 0.065 and 0.05 mm, respectively. The length and width of the sample were both 50 mm. Two conductor lines glued to the copper layers were used as the output lines, which were connected to a load resistor with a value of 1 or 10 GΩ in tests. A friction process between the left copper layer and the right PTFE layer brought triboelectric charges onto the two layers.

Figure 4b shows the sample test setups used in the experiments. In the tests, the right part in Figure 4a was fixed as the bottom, and the left part was placed above the PTFE layer, which brought the upper Cu layer onto the PTFE layer. Then, an up and downward motion of the upper Cu layer relative to the fixed PTFE layer was actuated by a motor. The motion of the Cu layer was a sinusoidal motion, and the displacement of the moving Cu layer relative to the PTFE layer was between ~0 and 10 mm. The motion frequency was about 1.23 Hz. When the upper Cu layer was in motion, the transfer charge *Q*_c_ accumulated in the TENG sample began to vary, as measured by a Keithley Model 6514 system electrometer (Tektronix, Shanghai, China), with the measured results recorded by a personal computer. These measured results are shown in Figure 4c,d. By comparison of Figure 4c,d, it was found that these curves exhibit the same trend as predicted by the theoretical work. With the increase in load resistance, the minimum value of transfer charges increased due to the obstruction of large resistance, and transient phenomenon became more evident, which supported the conclusions of theoretical prediction. It could also be predicted that a larger resistance should lead to a more apparent transient phenomenon.

## 5. Conclusions

In this work, the transient process of TENGs was studied based on an infinite-plate model and verified by experimental tests. A formula for predicting the duration of the transient process was derived via the Fourier expansion technique, which shows that the duration of the transient process is linearly proportional to the load resistance. It was found that during the transient process, the output performance of TENGs was much larger than that in the steady state, particularly for large load resistances. It was also found that during the transient process, the transfer charge gradually increased with time until reaching a maximum value, while the output current and power reduced with time. This phenomenon was explained by the obstruction of the resistor to the flow of transfer charges between the electrodes. The larger resistance would bring a greater obstruction. The results also revealed that during the transient process, the *Q*-*V* curves gradually transformed from an open loop to a closed loop. Furthermore, the transient phenomenon was clearly observed in experimental tests, which verified the theoretical prediction. According to these results, possible strategies based on the transient state of TENGs may be proposed to increase their output electric current and power, which are useful for power collection and self-powered sensor applications.

## Figures and Tables

**Figure 1 micromachines-16-01070-f001:**
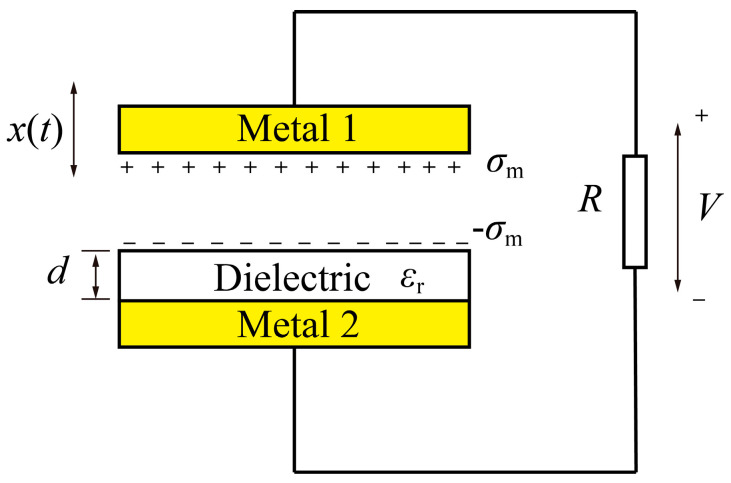
A typical model structure of the contact-mode TENGs.

**Figure 2 micromachines-16-01070-f002:**
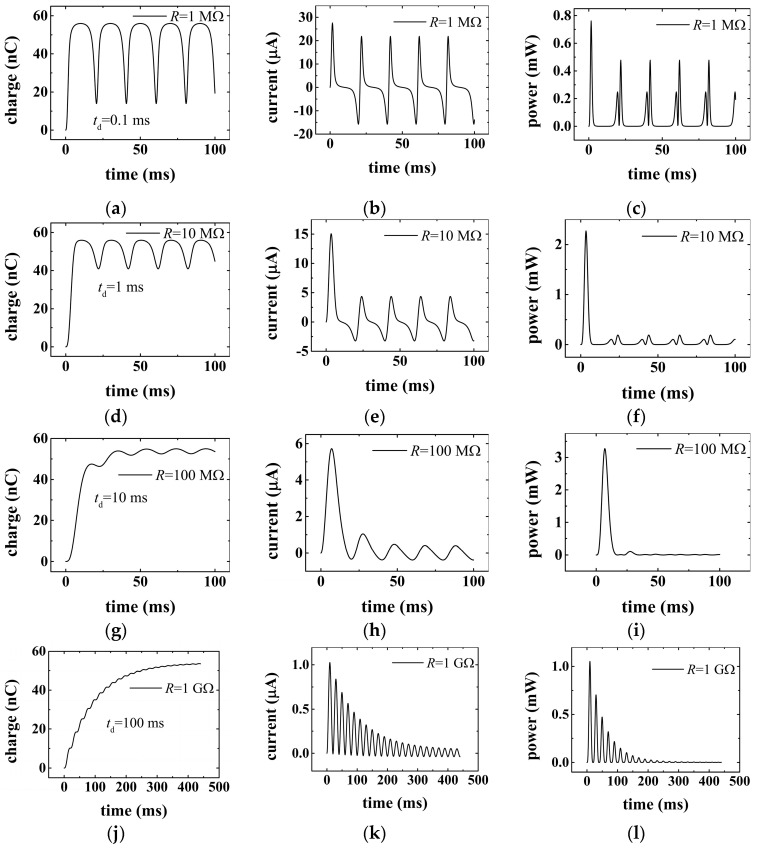
Demonstration of transient process of the contact-mode TENGs. (**a**–**c**) Variation in transfer charge, current, and power with time for load resistance R = 1 MΩ. (**d**–**f**) Variation in transfer charge, current, and power with time for load resistance R = 10 MΩ. (**g**–**i**) Variation in transfer charge, current, and power with time for load resistance R = 100 MΩ. (**j**–**l**) Variation in transfer charge, current, and power with time for load resistance R = 1 GΩ.

**Figure 3 micromachines-16-01070-f003:**
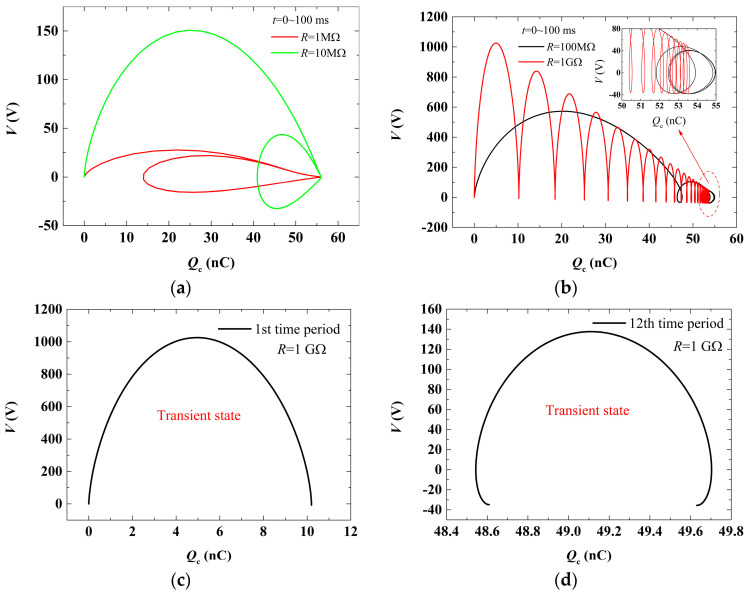
Variation in *Q*-*V* curve with time in the transient process of the TENG. (**a**) Variation in *Q*-*V* curves within a time range of 0~100 ms in the transient process for a small load resistance; (**b**) variation in *Q*-*V* curves within a time range of 0~100 ms in the transient process for a large load resistance; (**c**) the *Q*-*V* curve in the first time period for load value 1 GΩ; (**d**) the *Q*-*V* curve in the 12th time period for load value 1 GΩ; (**e**) the *Q*-*V* curve in the 22nd time period for load value 1 GΩ; (**f**) the *Q*-*V* curve in the 59th time period for load value 1 GΩ.

**Figure 4 micromachines-16-01070-f004:**
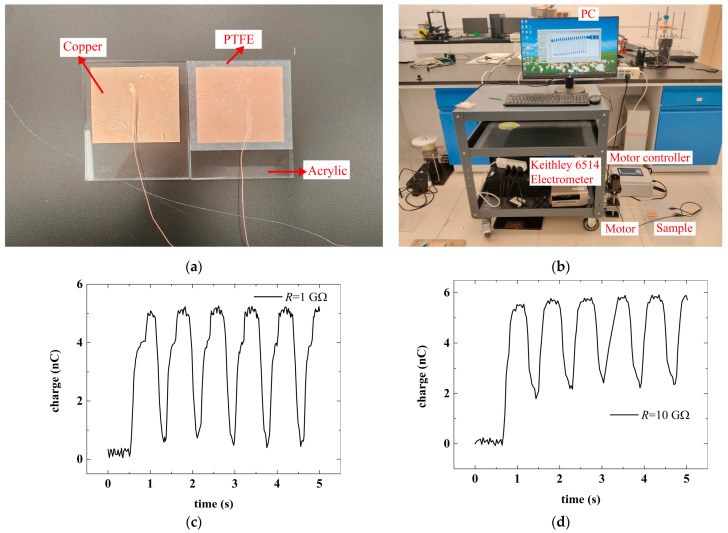
The experimental results of the dependence of transfer charge on time for load resistance of 1 or 10 GΩ. (**a**) The fabricated sample of contact-mode TENG; (**b**) the test setups for measuring the output performance of the TENG sample in the experiments; (**c**) the experimental results of the dependence of transfer charge on time for load resistance of 1 GΩ; (**d**) the experimental results of the dependence of transfer charge on time for load resistance of 10 GΩ.

**Table 1 micromachines-16-01070-t001:** Parameters utilized in the theoretical calculation for validation.

Parameter	Value
Dielectric thickness *d*	125 μm
Relative dielectric permittivity	3.4
Area size *S*	58 cm^2^
Triboelectric charge density $\sigma_{T}$	10 μC/m^2^
Maximum displacement x_m_	1 mm
Motion frequency *f*	50 Hz

## Data Availability

The original contributions presented in this study are included in the article. Further inquiries can be directed to the corresponding author.

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
