# Peer review of "A Study on the Transient Process of Contact-Mode Triboelectric Nanogenerators"

_micromachines, 2025, doi:10.3390/mi16091070_

Round 1

Reviewer 1 Report

Comments and Suggestions for Authors

Author formulated voltage differential equation during initial stage of contact-separate triboelectric generator. This work provide clear understanding of charge-voltage relation in terms of transient and steady state solutions. I recommend publish with few modifications.

  1. line 138, “The transition process duration calculated from Equation (9) is shown in Figure(2) for comparison with these curves.”, which has no meaning. Please correct figures(2) and provide correct comparison.
  2. Fig 4 has two experimental results depending on load resistance. Why author did not directly compare simulation and experimental result. Is there big difference between them? Please, provide analytic or simulation result corresponding experimental condition.

Author Response

We are grateful to the reviewer for the positive and constructive comments. Their insightful suggestions have provided us with valuable ideas to improve our manuscript. Accordingly, we have carefully revised the manuscript to address all the points raised. Our detailed point-by-point responses are included in the attached document.

Reviewer 2 Report

Comments and Suggestions for Authors

This work investigates the transient process of contact-mode TENGs using the infinite-plate model, supported by experimental validation. The results reveal that the electrical output of TENGs is higher during the transient phase than in the steady state. In this transient process, the transferred charge gradually increases to its maximum value, while the output current and power decrease. By applying Fourier expansion, the duration of the transient process was determined. Overall, this study highlights an intriguing transformation of the Q–V curve during the transient process. Furthermore, the phenomenon was experimentally verified using a contact-mode TENG fabricated from copper and PTFE films.

The manuscript might only be accepted after addressing the following comments:

Major comments:

  1. In section 2 (Theoretical Basis), the model structure in Fig.1 is not a Single electrode. It is mentioned in the caption of Fig.4a, please revise it.
  2. In Fig.2, explain the relationship between the external load resistance during the transition process of the contact-mode TENGs versus the transfer charge, current and power? Why as the increase of load resistance, the transfer charges are increase but it is vice versa for the current and power?
  3. Also, in Fig.2a-d-g-j, explain the reason of the shifting of charge waveform as the external resistance changes in the theoretical calculation. In this figure, the connection to higher load resistance provides the shrink curve than that of the lower load resistance, i.e., the peak-to-peak charge curves are smaller. Please clarify and add the explanation in the manuscript. The same explanation to that of current and power?
  4. The curves in Fig.2 and Fig.3 should be connected to the formula above, in each description related to the Figure, which formula is used should be mentioned.
  5. In Fig.4b, the wires are connected to acrylic is totally wrong. Please revise it.

Minor comments:

  1. In Fig.1, the thickness of dielectric material should be shown in the schematic, as well as the relative dielectric constant, external load, x(t).
  2. All figures should be provided with high resolution.
  3. Please review the caption in Fig.3.
  4. 4a should be recaptured with a better view, well arrangement and more professional style.
  5. It would be good if the authors mention which device is used to measure for the experimentally verified using a contact-mode TENG fabricated from copper and PTFE films.
Comments on the Quality of English Language

The English language must be improved.

Author Response

(The authors gave the same response as above.)
